# Aggregation-Induced Ignition of Near-Infrared Phosphorescence of Non-Symmetric [Pt(C^N*N’^C’)] Complex in Poly(caprolactone)-based Block Copolymer Micelles: Evaluating the Alternative Design of Near-Infrared Oxygen Biosensors

**DOI:** 10.3390/bios12090695

**Published:** 2022-08-28

**Authors:** Nina A. Zharskaia, Anastasia I. Solomatina, Yu-Chan Liao, Ekaterina E. Galenko, Alexander F. Khlebnikov, Pi-Tai Chou, Pavel S. Chelushkin, Sergey P. Tunik

**Affiliations:** 1Institute of Chemistry, St. Petersburg State University, Universitetskii Av., 26, 198504 St. Petersburg, Russia; 2Department of Chemistry, National Taiwan University, No. 1, Sec. 4, Roosevelt Rd., Taipei 10617, Taiwan

**Keywords:** Pt(II) complexes, aggregation-induced emission, polymer micelles, phosphorescence lifetime imaging, oxygen biosensors

## Abstract

In the present work, we described the preparation and characterization of the micelles based on amphiphilic poly(ε-caprolactone-*block*-ethylene glycol) block copolymer (**PCL-*b*-PEG**) loaded with non-symmetric [Pt(C^N*N’^C’)] complex (**Pt1**) (where C^N*N’^C’: 6-(phenyl(6-(thiophene-2-yl)pyridin-2-yl)amino)-2-(tyophene-2-yl)nicotinate). The obtained nanospecies displayed the ignition of near-infrared (NIR) phosphorescence upon an increase in the content of the platinum complexes in the micelles, which acted as the major emission component at 12 wt.% of **Pt1**. Emergence of the NIR band at 780 nm was also accompanied by a 3-fold growth of the quantum yield and an increase in the two-photon absorption cross-section that reached the value of 450 GM. Both effects are believed to be the result of progressive platinum complex aggregation inside hydrophobic poly(caprolactone) cores of block copolymer micelles, which has been ascribed to aggregation induced emission (AIE). The resulting phosphorescent (**Pt1@PCL-*b*-PEG**) micelles demonstrated pronounced sensitivity towards molecular oxygen, the key intracellular bioanalyte. The detailed photophysical analysis of the AIE phenomena revealed that the NIR emission most probably occurred due to the excimeric excited state of the ^3^MMLCT character. Evaluation of the **Pt1@PCL-*b*-PEG** efficacy as a lifetime intracellular oxygen biosensor carried out in CHO-K1 live cells demonstrated the linear response of the probe emission lifetime towards this analyte accompanied by a pronounced influence of serum albumin on the lifetime response. Nevertheless, **Pt1@PCL-*b*-PEG** can serve as a semi-quantitative lifetime oxygen nanosensor. The key result of this study consists of the demonstration of an alternative approach for the preparation of NIR biosensors by taking advantage of in situ generation of NIR emission due to the nanoconfined aggregation of Pt (II) complexes inside the micellar nanocarriers.

## 1. Introduction

Since the coining of the term “Aggregation-Induced Emission (AIE)” by B. Z. Tang’s group in 2001 [1], various aggregation-induced [2,3,4] or aggregation-enhanced [5] photophysical phenomena have been described. Such an intense focus on AIE and related phenomena stems from realizing the potential benefits of implementing AIE-based emitters (AIEgens) in applications utilizing high emitter concentrations, such as OLED technology [1], organelle-specific imaging and sensing [6], as well as photodynamic therapy [7]. The major advantage of these emitters is that contrary to conventional emitters that quench upon concentrating, AIEgens ignite or enhance their luminescence, two-photon emission, singlet oxygen production, etc. [2,3,4].

Because the very essence of AIE lies in the rigidification of the emitters’ microenvironment that restricts intramolecular motions and thus suppresses the rotovibrational relaxation of excited states [2,3,4], one can expect that organometallic compounds, in line with “classical” organic AIEgens, are also prone to exert AIE and related phenomena. Indeed, recent reports reviewed in [8] demonstrated that various (iridium(III) [9,10], ruthenium(II) [11,12], rhenium(I) [13,14,15], gold(I) [16], etc.) organometallic complexes also demonstrate AIE or related photophysical (e.g., increase in two-photon emission cross-sections [10]) and photochemical (e.g., enhancement of singlet oxygen generation [9,10]) behavior. In this context, square planar Pt(II) complexes are among the ‘brightest’ examples demonstrating AIE and related phenomena [5,17,18,19,20].

Organometallic AIEgenes based on Pt(II) complexes are unique objects because they combine high sensitivity to oxygen (which naturally stems from the triplet nature of their excited states effectively quenched by triplet molecular oxygen) with a pronounced bathochromic shift of luminescence due to the emergence of metal–metal-to-ligand charge-transfer (^3^MMLCT) states [5,17,18,19,20] arising as a result of intermolecular Pt…Pt interaction. The most important practical result of this combination of properties is that such AIEgenes can provide a new strategy for the design of NIR phosphorescent oxygen nanosensors. Indeed, despite the impressive progress in the field of phosphorescent O_2_ sensors such as the development of NIR O_2_ sensors based on single molecules protected by oligo(ethylene glycol) [21,22], dendonized poly(ethylene glycol) (Oxyphor 2P [23]), embedded in polymer nanoparticles (NanO2-IR [24]), or developing macroscopic sensors for in vivo [25] and inside 3D cell cultures [26], the main paradigm of their design (i.e., using increasingly sophisticated organometallic complexes that possess NIR emission in a monomeric state [27]) is still a challenging synthetic task. Alternative approaches for the generation of NIR phosphorescence are required, where shifting the phosphorescence emission to NIR due to AIE could provide the desired opportunity.

In view of the possible practical applications of organometallic AIEgenes as oxygen biosensors, their dispersion stability and compact packing are of significant importance. Generally, AIE can be spatially restricted to small (down to nanometer scale) volumes, which seems to be quite useful in numerous applications [7]. In particular, such a nano-confinement of AIE provides the possibility for the controlled design of nanoparticle-based oxygen biosensors, as was reported recently for the sensor prototype based on symmetrical [Pt(C^N*N^C)] complexes embedded into aminated polystyrene nanoparticles that demonstrated aggregation-enhanced dual emission in the green and deep red areas of the visible spectrum [5]. Notably, the dispersion stability of the polystyrene nanoparticles used in the above study was based on electrostatic repulsions of amino groups; this type of stabilization is not sufficient in aqueous saline media due to electrostatic screening by low molecular weight salts [28].

In our own research, we recently developed a series of Pt(II) complexes with non-symmetric C^N*N’^C’ ligands and found that one of these [Pt(C^N*N’^C’)] complexes (here and below referred to as **Pt1**; Figure 1) displays an orange emission in its monomeric form but demonstrates the appearance of AIE in the near-infrared (NIR) region [29]. Taking into account the fact that the NIR emission fits well into the so-called “biological transparency window” (a region from 650 to 900 nm where light penetrates into tissues much deeper compared to radiation of visible spectral interval [30]), we decided to evaluate the performance of this complex in oxygen sensing. To implement the idea of AIE nanoconfinement, we embedded the **Pt1** complex into block copolymer micelles prepared from poly(ε-caprolactone-*block*-ethylene glycol) (**PCL-*b*-PEG**; Figure 1). This amphiphilic diblock copolymer forms ’core–shell’ type micelles, which proved to be more promising nanocarriers compared to ’plain’ polymer nanoparticles and latexes due to their improved stability in saline aqueous dispersions [31]. The resulting **Pt1@PCL-*b*-PEG** nanosensor demonstrated prominent AIE-based NIR emission in combination with improved dispersion stability and an appreciable lifetime sensitivity to oxygen.

## 2. Materials and Methods

### 2.1. General Comments

All the basic laboratory procedures and measurement protocols are described in detail in Part 1 of the Appendix A. Additionally, experiments such as the estimation of micelles’ loading by **Pt1** with UV/Vis absorption spectroscopy, dynamic light scattering and photophysical experiments, as well as cell culture maintenance and a cell viability (MTT [32]) assay were performed using the protocols developed in our previous publication on phosphorescent micelles [31] with slight modifications. Hypoxia in cells was induced by blowing the nitrogen-air mixture at preset volume ratio above the cell monolayer for ca. 30 min [21,33]. The exact protocols used in the present study can also be found in Part 1 of the Appendix A.

### 2.2. Handling of **PCL-b-PEG** Block Copolymers and Preparation of Block Copolymer Micelles

The 2 samples of **PCL-*b*-PEG** used in the study were both purchased from “Sigma-Aldrich”, St. Louis, MO, USA (Product # 570303). A summary of the characterization of both samples by ^1^H NMR and GPC is presented in Appendix A.

Sample #1 was identical to that used in our previous work [31]. In the case of this sample, the protocol reported earlier [31] facilitated the reproducible preparation of almost transparent micellar dispersions without signs of precipitation for both ‘empty’ **PCL-*b*-PEG** micelles and those loaded by **Pt1** (**Pt1@PCL-*b*-PEG**), so we used this protocol for all the preparations based on sample #1.

Sample #2 was new; in the case of this sample, using of the same protocol [31] resulted in the formation of turbid micellar dispersions that yielded precipitates during preparative centrifugation for at least 15 min at 15,000 rpm. Consequently, before its further use in micelle preparation, sample #2 was ‘purified’ by performing the following cycles: first, micelles were prepared by the protocol reported earlier [31]; then, the dispersions were separated from the sedimentally unstable fraction by performing preparative centrifugation followed by freeze drying of the supernatant. The resulting lyophilizate was used as a starting material for the next cycle. After three cycles described above, the ‘purified’ version of **PCL-*b*-PEG** sample #2 was obtained with an overall yield of 80%. This sample formed almost transparent micellar dispersions.

In the case of the ‘purified’ version of **PCL-*b*-PEG** sample #2, use of the preparation protocol described earlier [31] for the formation of ‘loaded’ **Pt1@PCL-*b*-PEG** micelles led occasionally to parallel precipitation of **Pt1**, so in this case we used a slightly modified preparation protocol featuring a faster and more uniform addition of water via the syringe operated by a speed-adjustable stepping motor and an elevated temperature of the reaction mixture from ambient (ca. 25 °C) to 40 °C. The modified protocol consisted of the separate dissolving of both starting components (block copolymers and complex **Pt1**) in DMF to obtain stock solutions of 30 mg/mL for **PCL-*b*-PEG** of 5 mg/mL for **Pt1**, their subsequent mixing at appropriate weight proportions resulting in the **Pt1**:block copolymer mixtures with 0 (‘empty’ micelles), 2, 6, 9, or 12 wt.% of the complex **Pt1** at 40 °C followed by the addition of 3 volumes of type 1 water under vigorous stirring (1200 rpm) to form the micelles. The flow rate of water was adjusted to accomplish the addition of water within 15 min or less (typically, this required the flow rate of 0.2 mL/min) to avoid **Pt1** precipitation. Water was added via a 10 mL sterile disposable syringe operated by a speed-adjustable stepping motor (model: YH42BYGH60-401A, “Cnyoho”, China) to provide a steady flow of the liquid. The dispersions were then purified from DMF by performing dialysis using “Spectra/Por^®^4” dialysis tubes (“Scienova”, Germany) with a molecular weight cut-off of 12–14 kDa against type 1 water (5–7 water changes). The obtained micelle dispersions were transferred into pre-weighted 12 mL vials to evaluate mass concentration and stored in a fridge in the dark at 4 °C. Measured mass concentrations of the final dispersion were obtained by the weighting of lyophilizates obtained by performing freeze-drying of pre-weighted aliquots of the dispersions. Measured and evaluated concentrations of the dispersion agreed well within the experimental error (less than 12%).

### 2.3. Measurerents of Two-Photon Properties

The two-photon absorption (TPA) cross section of **Pt1@PCL-*b*-PEG** at 800 nm was measured by performing an open aperture Z-scan experiment, where a mode-locked Ti: sapphire laser (Tsunami, Spectra Physics) producing single Gaussian pulse (800 nm) was coupled to a regenerative amplifier to generate an approx. 180 fs and 1 mW pulse (760–840 nm, 1 kHz, Spitfire, Spectra Physics) as the excitation source. The pulse energy, after appropriate attenuation, was then reduced to ~1.2–1.8 μJ. After passing through an f = 7.5 cm lens, the laser beam was focused and passed through a 1.00 mm cell with solution (4 × 10^−4^–1 × 10^−3^ M), and the beam radius at the focal position was 0.1 cm. When the sample cell changed its position along the beam direction (z-axis), the transmitted laser beam from the sample cell was detected by a photodiode (PD-9, Ophir). The TPA-induced decrease in transmittance, T (*z*), can be expressed as Equations (1) and (2), and the TPA coefficient (*β*) can be obtained from experimental data by fitting Z-scan curves to Equations (1) and (2).
(1)$$T\left(z\right)=\overset{\infty }{\underset{0}{\sum }}\frac{{\left(-q\right)}^{n}}{{\left(n+1\right)}^{3/2}}$$
(2)$$q=\frac{\beta I_{0}L}{1+\frac{z^{2}}{z_{0}{}^{2}}},$$
where *n* is an integer number from 0 to ∞ and is truncated at *n* = 30, *L* is the sample length and *I*_0_ is the input intensity. *z*_0_ is the diffraction length of the incident beam (Rayleigh range). After obtaining the TPA coefficient (*β*; Equation (3)), the TPA cross section (*σ*_TPA_) can be deduced by using Equation (3), where *N_A_* is the Avogadro number, *d* is the concentration, *h* is the Plank constant, and *υ* is the frequency of the incident beam. The error of TPA measurement when utilizing the Z-scan method was in the range of <5% after five replicas.
(3)$$\beta =\frac{\sigma_{\mathrm{TPA}}N_{A}d\times 10^{-3}}{h\nu }$$

### 2.4. Confocal Microscopy and PLIM

Living cells were imaged by using a confocal inverted Nikon Eclipse Ti2 microscope (Nikon Corporation, Tokyo, Japan) with 60× oil immersion and 40× water immersion objectives. The images were recorded using the following standard settings: excitation at 405 nm and emission at 663–738 nm. Differential interference contrast (DIC) images were also recorded in addition to fluorescence microphotographs. Phosphorescence lifetime imaging microscopy (PLIM) of cells was carried out using the TCSPC DSC-120 module (Becker & Hickl GmbH, Berlin, Germany) integrated into the confocal device. A picosecond laser (405 nm) was used as an excitation source. The phosphorescence of the probe was recorded in two channels using a (1) 630/75 nm band pass filter and pinhole of 0.5, (2) 720/60 nm band pass filter and pinhole of 1.0. In normoxic conditions, the following PLIM acquisition parameters were used: frame time 20.82 s, pixel dwell time 78.90 µs, points number 1024, time per point 75.00 ns, time range of PLIM recording 76.80 µs, total acquisition time 100–130 s, and image resolution 512 × 512 pixels. In hypoxic conditions, the parameters were as follows: frame time 41.97 s, pixel dwell time 159.6 µs, points number 1024, time per point 150.00 ns, time range of PLIM recording 153.60 µs, total acquisition time 100–150 s, and image resolution 512 × 512 pixels. Oil immersion 60× objective with zoom 5.33 provided a scan area of ca. 0.05 mm × 0.05 mm. Water immersion 40× objective with zoom 5.33 provided a scan area of ca. 0.08 mm × 0.08 mm. Phosphorescence lifetime distributions were calculated using SPCImage 8.1 software (Becker & Hickl GmbH, Berlin, Germany). The decay curves were fit as triexponential decay with an average goodness of fit of 0.8 ≤ χ^2^ ≤ 1.1. The average number of photons under the decay curve were ≥5000 at binning of 7–8.

## 3. Results

### 3.1. Preparation and Structural Characterization of Phosphorescent Micelles

Two samples of **PCL-*b*-PEG** block copolymer were used in the study. Sample #1 was identical to that used in our previous work [31]. Its [PEG]:[PCL] ratio calculated from ^1^H NMR spectra (Appendix A) was 4.4:1.0, which exceeded the corresponding value expected from *M_n_* values of blocks (*M_n_* (PEG) = *M_n_* (PCL) = 5000; [PEG]:[PCL] = 2.6:1.0) provided by the manufacturer (Appendix A). The average molecular weight (*M_w_*) and dispersity (*Ð*) of this sample also slightly exceeded the corresponding values provided by the manufacturer, but were in reasonable agreement with these data (Appendix A). ‘Empty’ micelles from this sample were prepared by performing the protocol reported earlier [31]; these micellar dispersions were almost transparent and lacked any signs of precipitation; the corresponding hydrodynamic radii (*R_h_*) distribution obtained by DLS was reported to be almost unimodal with trace amounts of larger (60–200 nm) particles while the average *R_h_* value was reported to be 17.7 nm [31].

Sample #2 has not been reported previously. The ^1^H NMR spectrum of the starting polymer (“Sigma-Aldrich”, St. Louis, MO, USA; Product # 570303) was almost identical to that of sample #1 (Appendix A) but displayed a slightly lower [PEG]:[PCL] ratio (3.8:1.0) that indicates its lower hydrophilicity (i.e., lower PEG content) compared to sample #1. GPC revealed that sample #2 had a lower *M_w_* (*M_w_* (sample #2) = 10,800 compared to *M_w_* (sample #1) = 15,000; Appendix A) and higher dispersity (*Ð* (sample #2) = 1.41 compared to *Ð* (sample #1) = 1.33; Appendix A). Most probably, the lower [PEG]:[PCL] ratio (i.e., lower hydrophilicity) and higher dispersity were the major reasons for sample #2 to form substantially turbid micellar dispersions, which eventually yielded precipitates. Nevertheless, three repetitive cycles of precipitate separation (micelle preparation/precipitate separation by preparative centrifugation/freeze drying of the supernatant; see Section 2.2. in Materials and Methods for more details) allowed us to obtain the ‘purified’ version of **PCL-*b*-PEG** sample #2 with lower *Ð* = 1.31 and [PEG]:[PCL] = 4.5:1.0 (both these parameters of ‘purified’ sample #2 appeared to be very close to those of sample #1; Appendix A). This ‘purified’ sample #2 reproducibly formed almost transparent micellar dispersions. The corresponding *R_h_* distribution of micelles based on the ‘purified’ sample #2 obtained by DLS was almost unimodal with similar trace amounts of larger (100–200 nm) particles and had average *R_h_* values of 22 ± 2 nm (Figure 1A). Because *M_w_*, *Ð, R_h_* distributions, and average *R_h_* values of the sample #1 and the ‘purified’ sample #2 were essentially similar, both **PCL-*b*-PEG** samples were used in the study interchangeably.

Loading of **Pt1** into **PCL-*b*-PEG** micelles (i.e., formation of **Pt1@PCL-*b*-PEG** micelles) required a slight modification of the preparation protocol used previously [31] due to the pronounced tendency of **Pt1** to precipitate in the water:DMF mixtures even in the presence of block copolymer micelles. Nevertheless, the faster (accomplished in less than 15 min) and steadier addition of water (via the syringe operated by speed-adjustable stepping motor) into the reaction mixture at a slightly elevated temperature (40 °C) made possible the successful preparation of **Pt1@PCL-*b*-PEG** micelles. The resulting content of **Pt1** in the micelles was only slightly lower than that in the starting reaction mixture (Figure 1B). Investigation of **Pt1@PCL-*b*-PEG** micelles by DLS revealed that their *R_h_* distributions were almost unimodal (Figure 1A) with a minor peak of larger (of approx. 80–200 nm) particles and closely resembled the *R_h_* distribution of ‘empty’ micelles. The average *R_h_* values of **Pt1@PCL-*b*-PEG** micelles were 19 ± 2 and 18 ± 2 nm at 6.1 wt.% and 12.1 wt.% of **Pt1**, respectively. The ‘empty’ and ‘loaded’ micelles prepared via the modified protocol were stable during preparative centrifugation for 15 min at 20,000 rpm without any precipitation. Additionally, no phase separation or precipitation was observed at concentrations of up to 3 mg/mL, the upper concentration limit in stock solutions, as well as during dilution or storage at 4 °C for at least two weeks. Similar stability was observed in PBS solutions.

### 3.2. Photophysical Characterization of Phosphorescent Micelles

Photophysical characterization of **Pt1@PCL-*b*-PEG** started with the measurements of absorption, excitation and luminescence spectra of **Pt1@PCL-*b*-PEG** samples with **Pt1** loading ranging from 1.8 to 12.1 wt.%. Figure 2A,B demonstrate that the variation of the **Pt1** content in micelles does not affect the absorption and excitation spectra (see also Appendix A for more detailed representations of excitation spectra recorded at 605 and 780 nm, respectively). On the contrary, the corresponding luminescence spectra change drastically; in addition to the structured emission band of non-aggregated **Pt1** with the major peak at 605 nm, the increase in **Pt1** loading led to the appearance of a new emission band in the NIR range (Figure 2C). Moreover, the overall emission intensity increased with increased loading of **Pt1** into the micelles (Appendix A). Deconvolution of the spectral pattern (subtraction of monomolecular **Pt1** spectra in THF from those of **Pt1@PCL-*b*-PEG**) revealed that the profile of this NIR band does not depend on **Pt1** content and appears as a broad symmetrical band centered at ca. 790 nm with a half-width of ca. 180 nm (Figure 2D). It is also worth noting that the increase in the **Pt1** content resulted in a strong increase in the relative intensity of the NIR band and became dominant starting from approx. 8 wt.% of **Pt1** (Figure 2C). The quantum yield values measured in aerated aqueous dispersions of **Pt1@PCL-*b*-PEG** increased from 2% for 2.5 wt.% of **Pt1** loading to 6% for 12.4 wt.% of **Pt1** loading.

To understand the nature of the excited state, the nanosecond time-resolved measurement was performed by using TCSPC. As shown in Appendix A, a corresponding decay and rise of about 19.54 ns was detected upon monitoring the dual emission at 605 nm and 790 nm, respectively, for **Pt1@PCL-*b*-PEG** (2.3 wt.% of **Pt1**) in the aerated water. Therefore, the dual emission possesses a precursor–successor type of relationship whereby the NIR 790 nm emission (successor) originates from the relaxation of the 605 nm monomer emission (precursor). This viewpoint is also supported by the identical excitation spectrum between two monitored emission regions, e.g., 605 and 790 nm. Together with the concentration-dependent emission spectra, the 790 nm NIR emission was most plausibly assigned to the excimer emission of **Pt1** aggregates.

In our recent publication [29], two-photon absorption (σ_TPA_) and emission (σ_TPE_) cross-sections of **Pt1** in organic solvents were measured by a comparative method to produce the values of 20 and 10 GM, respectively. Since the observed aggregation can also enhance non-linear optical properties of **Pt1**, we measured the σ_TPA_ for **Pt1@PCL-*b*-PEG** micelles, too. As a result, the two-photon absorption cross section (σ_TPA_) of **Pt1@PCL-*b*-PEG** (2.3 wt.% of **Pt1**) deduced by performing the Z-scan method was 450 ± 22 GM (800 nm) (see Appendix A). This value significantly improved compared to the **Pt1** monomer (130 ± 10 GM at 800 nm) estimated by the same method.

Lifetime measurements for the **Pt1@PCL-*b*-PEG** micelles loaded with 2.3 and 12.1 wt.% of the platinum complex were also carried out in different media and at different partial pressures of oxygen, see Figure 3, Appendix A; Table 1, Appendix A. A detailed investigation of the emission decay curves collected at different detection wavelengths (605, 665, 725 nm, Appendix A) indicated that in the case of data acquisition with reliable statistics (10^4^ photon counts) the biexponential fitting provides the following results: the lifetime of the dominant exponent (*τ*_major_) remains unchanged within the experimental uncertainty (*τ*_major_ = 0.71 ± 0.02 

s for **Pt1@PCL-*b*-PEG** loaded by 12.1 wt.% of **Pt1** in aerated water dispersion at 37.6 °C), and its contribution in the decay varies from 92 to 95% (Appendix A). These observations indicate that for the **Pt1@PCL-*b*-PEG** samples containing 12.1 wt.% of the complex (1) the emission decay curves in water and PBS (Appendix A) may be successfully fitted with the monoexponential function (Table 1 and Appendix A) and (2) lifetime measurements may be carried out using any of these detection wavelengths due to the major contribution of the NIR emission band to the decay independent of the wavelength choice. Lifetimes data at different O_2_ partial pressures were used to construct Stern–Volmer plots for **Pt1@PCL-*b*-PEG** dispersions (with **Pt1** loading ranging from 2.3 to 12.1 wt.%) in water and PBS at 37.6 °C (Figure 3A). Lifetime sensitivity to oxygen estimated as the τ_0_/τ_160_ ratio (where τ_0_ refers to 0 mmHg O_2_, τ_160_ to 160 mmHg O_2_) varied from 3.0 for the dispersions with 2.3 wt.% of **Pt1** (τ_160_ = 1.30 ± 0.07 

s; τ_0_ = 3.9 ± 0. 2 μs) to approx. 1.5 for that with 12.1 wt.% of **Pt1** (τ_160_ = 0.60 ± 0.03 μs; τ_0_ = 0.92 ± 0.05 μs), see Figure 3B and Table 1. The corresponding Stern–Volmer plots were linear for all micellar compositions (Figure 3A). Stern–Volmer constants (*K_SV_*) display values from 0.0033 ± 0.0002 mmHg^−1^ (**Pt1** loading: 12.1 wt.%) to 0.0124 ± 0.0001 mmHg^−1^ (**Pt1** loading: 2.3 wt.%) in Table 1. *K_SV_* values strongly depend on **Pt1** loading and level off above the compositions of 9 wt.% **Pt1**, whereas quenching constants, *K_Q_* do not depend on the **Pt1** content and equal to 0.0034 ± 0.0002 μs^−1^mmHg^−1^ for all studied micelles (Table 1).

We also studied the effects of the other media variables (pH, temperature, ionic strength, presence of H_2_O_2_ and bovine serum albumin) as well as the influence of the probe concentration on the lifetimes of aerated dispersions of **Pt1@PCL-*b*-PEG** (2.3 and 12.1 wt.% of **Pt1**), see Table 2. Lifetimes of these chromophores did not depend on the probe concentration, pH, ionic strength, and H_2_O_2_ (Table 2) but proved to be sensitive to the **Pt1** content in the micelles, temperature, and to the presence of bovine serum albumin (BSA). In the case of BSA, the decay curves became non-monoexponential (Table 2), and mean lifetime values were more than 1.5 times higher compared to those in water and PBS.

Further analysis of lifetimes was performed in complicated physiological media, namely, 85 vol.% fetal bovine serum, FBS; 85 vol.% calf serum, CS; 42.5 mg/mL BSA (this concentration is an equivalent of protein concentration in 85 vol.% serum). In these cases, lifetimes depended strongly on media composition and grew up to 4–5 μs in aerated and 8–10 μs in deaerated conditions with rather high media-to-media variations (Appendix A). Notably, the relative contribution of the NIR band substantially decreased (Appendix A) in the FBS and BSA solution of the equivalent concentration (42.5 mg/mL) while smaller concentrations of BSA (0.25 mg/mL) and variations in ionic strength (PBS vs. water) did not influence the luminescence spectrum of **Pt1@PCL-*b*-PEG** (Appendix A).

The biologically relevant calibration of **Pt1@PCL-*b*-PEG** oxygen sensors vs. oxygen partial pressure was performed using **Pt1@PCL-*b*-PEG** dispersions with 12.1 wt.% **Pt1** in 85 vol.% fetal bovine serum (FBS) at 37 °C and a PLIM setup. The sensor signal was collected using two channels, namely red (595–665 nm) and NIR (690–750 nm). The decay curves in this case were essentially non-monoexponential (Appendix A) and treatment of all the decay curves was performed by implementing triexponential fitting. The resulting Stern–Volmer plots collected from both channels were linear but did not overlay each other, see Figure 4A. In addition, the mean lifetimes of the same probe measured in cuvette in different media at a detection wavelength of 605 nm were overlaid for comparison. These data demonstrate that mean lifetime values measured using the PLIM setup were similar to their counterparts measured in cuvette. The resulting Stern–Volmer plots were linear and provided τ_0_ = 9.7 ± 0.5 μs and *K_SV_* = 0.0093 ± 0.0009 mmHg^−1^ for the case of the red channel and τ_0_ = 5.9 ± 0.3 μs and *K_SV_* = 0.0090 ± 0.0006 mmHg^−1^ for the case of the NIR channel.

### 3.3. In Vitro Investigation of **Pt1@PCL-b-PEG** Oxygen Probes Inside CHO-K1 Cells

**Pt1@PCL-*b*-PEG** micellar dispersions of various compositions were first evaluated for their toxicity towards the CHO-K1 cell line by performing an MTT assay. The phosphorescent micelles displayed only slight cytotoxicity that was comparable to that of ‘empty’ **PCL-*b*-PEG** micelles; in all cases, the cell viability was more than 80% for up to concentrations of 0.3 mg/mL (Figure 4B).

Further, we assessed the phosphorescence lifetimes of **Pt1@PCL-*b*-PEG** (12.1 wt.%) micellar dispersions in the CHO-K1 cell line under normoxic and hypoxic conditions by phosphorescence lifetime microscopy (PLIM) combined with conventional confocal microscopy (Figure 5). Analogously to Stern–Volmer plots, PLIM data were recorded using the same two acquisition channels. The living CHO-K1 cells were incubated with probe concentrations of 0.3 mg/mL for 24 h (cell viability at these conditions is more than 80%). Confocal microphotographs revealed complicated intracellular distribution including at least the following two major patterns: diffuse staining of the cytosol and dot-like inclusions (Figure 5A, left column). The same sample areas were then scanned in the PLIM mode to provide the lifetime maps shown in Figure 5A, middle and right columns. Lifetimes of the probe were within the interval from 3.5 to 6 μs in the case of normoxia with an average value of 4.0 μs, and from 6 to 10 μs in hypoxia with an average value of 7–8 μs (Figure 5B). The lifetime distributions in both cases are rather wide, demonstrating broad lifetime variations across the same sample.

Finally, investigation of the influence of incubation period (30 min, 2, 5, and 24 h) on the phosphorescence lifetimes of **Pt1@PCL-*b*-PEG** (12.1 wt.%) micellar dispersions was performed on the same CHO-K1 cell line under normoxic conditions by PLIM combined with conventional confocal microscopy (Figure 6). Confocal microscopy revealed that the probe’s signal became strong enough to perform PLIM even after 30 min incubation. PLIM data demonstrated a detectable shift in the ‘averaged’ lifetime (i.e., value averaged from lifetime distributions across the image) following the increase in incubation time from 3 to 4 μs at 30 min and at 24 h incubation, respectively.

## 4. Discussion

The present work reported on the aggregation-induced ignition of NIR phosphorescence in **Pt1@PCL-*b*-PEG** micelles loaded with the non-symmetric platinum (II) complex **Pt1** (Figure 1), which was recently reported by our group [29], together with the evaluation of this system as an intracellular phosphorescent oxygen probe. The micelles were prepared in a similar way to that described in our previous work [31] with slight modifications intended to suppress the precipitation of **Pt1** in the course of micelle formation. The non-emissive ‘empty’ **PCL-*b*-PEG** micelles were also prepared as the ‘control’ system that does not contain the platinum emitter.

The composition of micelles was found to be close to that of the starting reaction mixture (Figure 1B) though it was slightly lower than can be explained by the residual precipitation of **Pt1** even under optimized conditions. Nevertheless, the proposed preparation protocol facilitates the loading of **PCL-*b*-PEG** micelles by **Pt1** up to at least 12 wt.% which is in good agreement with our recent report on the high loading capacity of these micelles towards similar organometallic complexes [31]. The ‘loaded’ micelles revealed almost unimodal size distributions (Figure 1A) with the average hydrodynamic radius (*R_h_*) of **Pt1@PCL-*b*-PEG**, which did not exceed 19 nm for all the compositions studied and is close to the value obtained for the ‘empty’ micelles (22 nm) as well as to that reported earlier (17.7 nm [31]). Intensity weighted *R_h_* distributions also indicated a non-negligible contribution from larger particles, with an *R_h_* of ca. 80–200 nm, which presumably represent aggregates of micelles [31]). However, our previous evaluation of weight fractions of aggregates in similar phosphorescent micelles demonstrated that the comparable contribution of larger particles into light scattering is a result of substantially higher sensitivity of light scattering to the particle size and corresponds to trace amounts of aggregates [31]. Due to their small sizes, the resulting phosphorescent micelles appeared to be quite stable towards precipitation (*vide supra*, Section 3.1.) not only in water but also in PBS, and this behavior was superior compared to conventional polymer nanoparticles stabilized by electrostatic repulsion [28]. Consequently, the obtained **Pt1@PCL-*b*-PEG** polymer nanoparticles look very promising as biocompatible carriers for luminescent emitters.

The major photophysical effect observed for **PCL-*b*-PEG** micelles loaded with the mononuclear platinum emitters consists of the ignition of NIR luminescence. Figure 2C and Appendix A demonstrate emergence and stepwise growth of a broad luminescence band centered at ca. 790 nm with the increase in **Pt1** content in the micelles from 1.8 to 12.1 wt.% accompanied by a 3-fold increase in quantum yield values (from 2% for 2.5 wt.% of **Pt1** loading to 6% for 12.4 wt.% of **Pt1** loading). Similar effects were observed earlier for the analogous system (symmetric [Pt (C^N*N^C)] complex embedded into aminated polystyrene nanoparticles [5]) but in that case only deep-red emission was achieved. Based on the recent literature data [5,17,18,19,20] it is possible to assign the observed NIR luminescence to aggregation-induced emission. The aggregation of monomeric **Pt1** molecules in the micelles core is evidently the most natural process for the hydrophobic platinum complex in a hydrophilic media that was previously found in closely analogous systems, e.g., in the mixed (NCMe-H_2_O and THF-H_2_O) solvents [19,29], containing water and hydrophobic organic components. The NIR band profiles obtained by deconvolution of experimental emission spectra are nearly identical for all micelles compositions from 1.8 to 12.1 wt.% (Figure 2D) that points to the essentially similar nature of NIR emissive chromophores independent of the **Pt1** content in the micelles. The spectra of the obtained species differed only in the relative emission intensity from monomeric complexes and aggregated emitters that is a result of the concentration driven equilibria of the aggregates formation from mononuclear complexes n***Pt1** ↔ [**Pt1**]_n_. It was found earlier that even the formation of dimers (2***Pt1** ↔ [**Pt1**]_2_) with short Pt…Pt contacts completely changes the character of the emissive excited state. The dimers with metallophilic bonding either in the ground or in excited state are responsible for the red shift of emission compared to the starting chromophore and crucial changes in the character of emissive excited states [19,29]. It is worth noting that, contrary to the emission spectra, the excitation and normalized absorption spectra are almost independent of the extent of **Pt1** loading and the detection wavelength (Figure 2A,B and Appendix A). These findings indicate that the observed NIR emission most probably occurs from the excimer with a metal–metal-to-ligand charge-transfer (^3^MMLCT) character [34,35] rather than from the ground-state of the dimers or oligomers with the Pt…Pt bonding [5].

Time-resolved photophysical studies of the **Pt1@PCL-*b*-PEG** micelles with the lowest (2.3 wt.% of **Pt1**; Appendix A) and highest (12.1 wt.% of **Pt1**; Appendix A) emitter loadings were performed at the detection wavelength of 605 nm as the monomolecular and aggregated chromophores displayed a substantial contribution in the emission decay at this wavelength. This choice looks natural for the probe with a low **Pt1** content, where the observed emission was mainly determined by the monomolecular chromophore. Unfortunately, the TCSPC detector sensitivity at 780 nm was too weak to obtain reliable photon counting statistics for both types of species at this wavelength. However, the measurements for the probes with 12.1 wt.% of **Pt1** at intermediate wavelengths (665 and 725 nm), *vide supra* (Results Section)*,* for which the detector sensitivity was acceptable, revealed that even in the case of biexponential fitting, the contributions of the major decay component was 92–95% with the lifetime values independent of the detection wavelength (Appendix A). Taking into account these observations, we can conclude that the decays obtained at 605 nm for water and PBS dispersions were monoexponential, and the minor components can be neglected without a loss of valuable information. Thus, the resulting decay curves were satisfactorily fitted using the monoexponential function in the case of aqueous solutions (water, PBS) but the triple exponential treatment was necessary to fit the decay curves in the media containing biomacromolecules, e.g., bovine serum albumin (BSA) (Appendix A). Triexponential fitting was chosen because of the substantial non-monoexponential character of the corresponding decay curves (Appendix A) that could not fit by biexponential fitting in most cases. The physical rationale of such a sophisticated treatment lies in our hypothesis regarding the non-negligible interaction of **Pt1@PCL-*b*-PEG** with the serum components, predominantly, with BSA, *vide infra*.

The lifetimes of **Pt1@PCL-*b*-PEG** in water and PBS demonstrated substantial dependence on the partial pressure of oxygen (Figure 3B; Table 1). The ratio of lifetimes in degassed and aerated dispersions (τ_0_/τ_160_) varied from 1.5 in the case of 12.1 wt.% of **Pt1** to 3.0 in the case of 2.3 wt.% of **Pt1**. Stern–Volmer plots of **Pt1@PCL-*b*-PEG** (2.3 to 12.1 wt.%) luminescence recorded in aqueous dispersions (in both water and PBS) revealed the linear dependence of reciprocal lifetime on the partial pressure of molecular oxygen in solution and insensitivity of the sensory response towards the presence of low-molecular electrolytes (PBS vs. water), Figure 3A. The absolute values of lifetimes display rather pronounced dependence on the extent of **Pt1** loading, which is evidently related to transformation of emissive centers upon an increase in the **Pt1** content in the micelles, from predominantly mononuclear emitters to the aggregates with Pt…Pt bonding. At higher (>9 wt.%) **Pt1** loadings (Figure 3B), the lifetime stay nearly constant which indicates a major contribution from the aggregated emitters. Moreover, the combination of the independence of lifetime values as well as their sensitivities to oxygen at high **Pt1** (>9 wt.%) loadings with the independence of lifetime from detection wavelength at 12.1 wt.% **Pt1** loading for the case of 12.1 wt.% of **Pt1** suggests that, at least at this composition, the resulting phosphorescent micelles demonstrate preferential emission from aggregates at all detection wavelengths (including that of 605 nm). This situation becomes even more pronounced in aerated media, since under these conditions the emission of non-aggregated **Pt1** is strongly quenched due to higher sensitivity to oxygen. In this context, it is reasonable to focus on 12.1 wt.% **Pt1** loading during the evaluation of **Pt1@PCL-*b*-PEG** as an oxygen sensor (the last part of the study).

To elucidate lifetime cross-sensitivity to various biasing parameters, we measured the lifetimes of **Pt1@PCL-*b*-PEG** in aerated PBS at different pH ranging from 5.8 to 8.1, at different sensor concentrations (0.2 vs. 0.4 mg/mL) and two temperatures (25.6 °C vs. 37.6 °C) as well as in the presence of reactive oxygen species (100 nm H_2_O_2_, the upper limit of physiologically relevant H_2_O_2_ content) and BSA. The results summarized in Table 2 demonstrate that temperature and the presence of BSA are two microenvironmental parameters that, in addition to the extent of **Pt1** loading, substantially affect lifetime characteristics. The temperature effect is very well known, is not unexpectable and can be corrected by performing calibrations and PLIM experiments at the same temperature. Using more complicated biological fluids such as FBS or calf serum results in even more complex decay curves (Appendix A) that can be fit by triexponential fitting, *vide supra*. The resulting mean lifetimes (Appendix A) were substantially higher compared to those measured in water or PBS, as well as in 0.25 mg/mL BSA (Table 2). One can assume that the major component affecting lifetimes in these media are proteins (predominantly, BSA), and this hypothesis is supported by deep analogies in both emission spectra (Appendix A) and lifetimes (Appendix A) observed in BSA solutions of a high concentration (42.5 mg/mL) which is equivalent to this protein content in the 85 vol.% serum used in the present study.

Such an unexpected behavior of phosphorescent micelles in serum can be explained by the influence of serum proteins, especially albumin, on the integrity of the micelles. Indeed, BSA is a transport protein featuring high solubilization efficiency towards hydrophobic molecules [36]. At relatively low protein concentrations, the micelles rather effectively protect **Pt1** from interactions with proteins; however, at extremely high protein concentrations, it seems that BSA can compete with micelles for binding with **Pt1** and solubilize part of this complex by the formation of non-covalent adducts [37,38,39]. This can result in the appearance of a new form of the emitter (non-covalent adducts **Pt1-BSA**) featuring its own decay profile. Such an interaction further complicates the lifetime decay profile leading to the necessity of multiexponential fitting specific for serum and BSA solutions (Appendix A). The hypothesis of the dominant impact of BSA on lifetimes of phosphorescent micelles is corroborated by the fact that in BSA solutions of a high concentration (42.5 mg/mL) the spectral profile (Appendix A) and lifetime (Appendix A) changes are almost the same as in the FBS.

The major practically important consequence of the above findings is that the application of the sensor systems of this type in quantitative O_2_ sensing requires obtaining the calibration curves at a fixed temperature (equal to that intended for further sensing experiments) and in the model media with the composition as close as possible to that of cell cytosol. Recently, we found that FBS is a relevant model media mimicking intracellular environment [22]. Hence, we performed the calibration of **Pt1@PCL-*b*-PEG** (12.1 wt.%) in FBS at 37 °C at 0.3 mg/mL of the probe (Figure 4A) using the PLIM instrument. The resulting Stern–Volmer plot is linear (Figure 4A), and the absolute lifetime values are close to those measured in cuvette in FBS and 42.5 mg/mL BSA (Figure 4A).

MTT tests showed that the **Pt1@PCL-*b*-PEG** probe is not cytotoxic towards CHO-K1 cells if incubated for 24 h at concentrations of up to 0.3 mg/mL (Figure 4B). We thus evaluated the oxygen sensing performance of the **Pt1@PCL-*b*-PEG** probe at this concentration and incubation time by carrying out PLIM on CHO-K1 cells under the conditions of normoxia and hypoxia. The PLIM results demonstrate significant lifetime sensitivity of the **Pt1@PCL-*b*-PEG** probe towards O_2_ partial pressure comparable to other NIR O_2_ probes [22] in both red (595–665 nm) and NIR (690–750) channels (Figure 5). This level of sensitivity allows for the unambiguous detection of the oxygenation state of cells and differentiation of hypoxia vs. normoxia (the same result was achieved for lower **Pt1** loadings, see Appendix A). Moreover, the range of lifetimes variation fits well with the region observed in the Stern–Volmer plots that provides at least semi-quantitative evaluation of O_2_ partial pressures from PLIM data.

It is worth noting that **Pt1@PCL-*b*-PEG** rapidly accumulates inside the cells and distributes throughout the cytoplasm, and 30 min incubation is sufficient to obtain luminescence signal strong enough to run PLIM experiment (Figure 6). Dependence of ‘averaged’ lifetime (i.e., the value averaged from the lifetime distributions across the image) on incubation time demonstrates the progressive increase in this parameter during the incubation. The observed increase in lifetime upon increasing the incubation time could be the result of (at least a partial) interaction of **Pt1@PCL-*b*-PEG** with serum proteins that was shown to result in a lifetime increase, *vide supra*. Alternatively, the observed effect can correlate with the observed accumulation of dot-like inclusions inside cells that significantly contribute to the overall phosphorescence signal after 24 h incubation (Figure 6). Obviously, the unambiguous interpretation of the observed effect requires a much deeper evaluation of the intracellular fate of the **Pt1@PCL-*b*-PEG** probe and is well beyond the scope of the present study.

## 5. Conclusions

In conclusion, the present work described the first example of the preparation of biocompatible NIR emitters by using the effect of nanoconfined AIE generation from the non-symmetric [Pt (C^N*N’^C’)] complex in block copolymer micelles (**PCL-*b*-PEG**). The observed aggregation-induced NIR phosphorescence becomes more prominent at increased **Pt1** loading into the micelles and features a high two-photon absorption cross-section (450 GM). Both effects are believed to be the result of progressive **Pt1** aggregation inside the hydrophobic cores of the **PCL-*b*-PEG** micelles. Compared to the closest literature analog (symmetric [Pt (C^N*N^C)] complex embedded into aminated polystyrene nanoparticles [5]), the present **Pt1@PCL-*b*-PEG** O_2_ nanoprobe demonstrated improved stability in aqueous physiological dispersions combined with bright NIR phosphorescence retaining pronounced sensitivity towards molecular oxygen. The detailed time-resolved photophysical analysis of NIR phosphorescence in nanosecond interval revealed that NIR emission results from the excimeric excited state of the ^3^MMLCT character. Evaluation of the **Pt1@PCL-*b*-PEG** efficacy as a lifetime intracellular oxygen biosensor showed that the probe displays a linear response of the emission lifetime onto variations in the concentration of oxygen which, however, is compromised by a pronounced influence of protein components of the physiological media. Therefore, **Pt1@PCL-*b*-PEG** can serve only as a semi-quantitative lifetime O_2_ nanosensor. Nevertheless, the present study proved the applicability of an alternative approach for creating NIR O_2_ biosensors that avoids the sophisticated synthesis of NIR emitters by using the aggregation-induced ignition of NIR phosphorescence instead. Further improvement of the present sensor prototype can be achieved by the enhancement of its insensitivity to serum components, for example, by cross-linking the micelles.

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
