# Peer review of "Aggregation-Induced Ignition of Near-Infrared Phosphorescence of Non-Symmetric [Pt(C^N*N’^C’)] Complex in Poly(caprolactone)-based Block Copolymer Micelles: Evaluating the Alternative Design of Near-Infrared Oxygen Biosensors"

_biosensors, 2022, doi:10.3390/bios12090695_

Round 1

Reviewer 1 Report

In this manuscript, the authors report the preparation and characterization of the micelles based on amphiphilic poly(e-caprolactone-block-ethylene glycol) block copolymer (PCL-b-PEG) loaded with non-symmetric [Pt(C^N*N’^C’)] complex (Pt1) and its application to bioimaging.

 The experiments have been well performed.  The manuscript can be publishable in Biosensors in the current form.

Author Response

Reply to the Reviewer 1

Comments and Suggestions for Authors

In this manuscript, the authors report the preparation and characterization of the micelles based on amphiphilic poly(e-caprolactone-block-ethylene glycol) block copolymer (PCL-b-PEG) loaded with non-symmetric [Pt(C^N*N’^C’)] complex (Pt1) and its application to bioimaging.

The experiments have been well performed. The manuscript can be publishable in Biosensors in the current form.

Response: We would like to thank the Reviewer’s efforts in reviewing the manuscript and high overall evaluation of our work. We also would like to stress that, following other Reviewers’ suggestions and recommendations, we have substantially modified the manuscript by rewriting the “Introduction” section and other minor parts that have corresponding changes, moving several parts of “Materials and Methods” section into Supplementary Information, and adding some new experimental data (non-normalized emission spectra and estimations of quantum yields as well as description of nanoparticles stability) that do not change the major conclusions but provide in-depth proof of AIE phenomenon and information on the nanoparticles stability. We hope that these changes have further improved the manuscript.

Reviewer 2 Report

In this manuscript (biosensors-1852722), the authors reported a NIR oxygen biosensors based on Pt(II) complex (Pt1)@block copolymer. In your previous work, AIE property of Pt1 in the NIR region had been researched. The quenching of phosphorescence by O2 has been widely known. Moreover, the problems and challenges of related areas had not been discussed in Introduction, in order that innovation point of this work cannot be clarified. Up to now, there have been some studies about phosphorescence-based biosensors for monitoring oxygen in cell (Biosensors and Bioelectronics, 2019, 123, 131-140; Scientific Reports, 2017, 7, 8255). Although the authors' system seems to have some differences, I feel the originality is limited. Therefore, the manuscript is not suitable for the journal. The specific comments are as follows:

1. Introduction should be reorganized to highlight the novelty of this work.

2. Considering the NIR phosphorescence, in vivo O2 assay should be done. In addition, safety of Pt1@block copolymer should be studied.

3. AIE property of Pt1@block copolymer should be proved.

Author Response

Reply to the Reviewer 2

Comments and Suggestions for Authors

In this manuscript (biosensors-1852722), the authors reported a NIR oxygen biosensors based on Pt(II) complex (Pt1)@block copolymer. In your previous work, AIE property of Pt1 in the NIR region had been researched. The quenching of phosphorescence by O2 has been widely known. Moreover, the problems and challenges of related areas had not been discussed in Introduction, in order that innovation point of this work cannot be clarified. Up to now, there have been some studies about phosphorescence-based biosensors for monitoring oxygen in cell (Biosensors and Bioelectronics, 2019, 123, 131-140; Scientific Reports, 2017, 7, 8255). Although the authors' system seems to have some differences, I feel the originality is limited. Therefore, the manuscript is not suitable for the journal.

Response: We agree with the Reviewer that the originality of the work was insufficiently addressed, and this deficiency of the manuscript is the major reason for the reviewer’s decision regarding the suitability of the presented manuscript for the Biosensors journal. We have addressed our inadvertence by the following steps: i) we have rewritten the “Introduction” section to highlight more strictly the novelty of our work (for more detail, please see our response to the specific comment #1 below); ii) we have also changed to some extent the final part of “Discussion” and “Conclusions” sections to further stress the novelty of the presented work; iii) both of the above changes resulted in the expansion of the list of cited literature (new references: [21-28], including (but not limited to) the papers mentioned by the Reviewer).

We believe that the clearer justification for both goal and novelty of the work should convince the Reviewer the suitability of this manuscript for publishing in Biosensors.

The specific comments are as follows:

  1. Introduction should be reorganized to highlight the novelty of this work.

Response: Done. A new paragraph describing state-of-the-art research and the major challenges in the area of developing NIR oxygen sensors has been added into the “Introduction” section (paragraph #3 and, partially, #4; Page 2):

“Organometallic AIEgenes based on Pt(II) complexes are unique objects because they combine high sensitivity to oxygen (which naturally stems from the triplet nature of their excited states effectively quenched by triplet molecular oxygen) with a pronounced bathochromic shift of luminescence due to emergence of metal–metal-to-ligand charge-transfer (3MMLCT) states [5,17–20] arising as a result of intermolecular Pt…Pt interaction. The most important practical result of this combination of properties is that such AIEgenes can provide a new strategy for the design of NIR phosphorescent oxygen nanosensors. Indeed, despite the impressive progress in the field of phosphorescent O2 sensors such as development of NIR O2 sensors based on single molecules protected by oligo(elthylene glycol) [21,22], dendonized poly(elthylene glycol) (Oxyphor 2P [23]), embedded in polymer nanoparticles (NanO2-IR [24]), or developing macroscopic sensors for in vivo [25] and inside 3D cell cultures [26], the main paradigm of their design (i.e. using of more and more sophisticated organometallic complexes that possess NIR emission in monomeric state [27]) is still a challenging synthetic task. Alternative approaches for the generation of NIR phosphorescence are required, where shifting the phosphorescence emission into NIR due to AIE could provide the desired opportunity.

In view of possible practical applications of organometallic AIEgenes as oxygen biosensors, their dispersion stability and compact packing are of significant importance. …    … Notably, the dispersion stability of the polystyrene nanoparticles used in the above study is based on electrostatic repulsions of amino groups; this type of stabilization is not sufficient in aqueous saline media due to electrostatic screening by low molecular weight salts [28].”

Further, the novelty of the work was additionally highlighted in the “Discussion” section (Page 13):

“Due to their small sizes, the resulting phosphorescent micelles appeared to be quite stable towards precipitation (vide supra, Section 3.1.) not only in water but also in PBS, and this behavior is superior compared to conventional polymer nanoparticles stabilized by electrostatic repulsion [28].”

“Figures 2, C and S4 demonstrate emergence and stepwise growth of a broad luminescence band centered at ca. 790 nm with the increase in Pt1 content in the micelles from 1.8 to 12.1 wt.% accompanied by 3-fold increase in quantum yield values (from 2 % for 2.5 wt.% of Pt1 loading to 6 % for 12.4 wt.% of Pt1 loading). Similar effects were observed earlier for the analogous system (symmetric [Pt(C^N*N^C)] complex embedded into aminated polystyrene nanoparticles [5]) but in that case only deep-red emission could be achieved.”,

In the  “Conclusions” section (Page 16):

“Compared to the closest literature analog (symmetric [Pt(C^N*N^C)] complex embedded into aminated polystyrene nanoparticles [5]), the present Pt1@PCL-b-PEG O2 nanoprobe demonstrates improved stability in aqueous physiological dispersions combined with bright NIR phosphorescence retaining pronounced sensitivity towards molecular oxygen.”

  1. Considering the NIR phosphorescence, in vivo O2 assay should be done.

In addition, safety of Pt1@block copolymer should be studied.

Response: Both suggestions provided by the Reviewer, namely, in vivo PLIM sensing as well as safety experiments (that can be assessed by acute toxicity tests as well as by tracking the effects caused by prolonged exposures) will require the experiments with animals. At present stage of development of this study, this step does not seem to be reasonable because the sensory performance of the probe in its current form is out of optimal. Further probe’s optimization including both the probe and block copolymer variations are far beyond the scope of the present work because its major goal is to provide the first demonstration of applicability of polymer micelles as nanocontainers for AIEgens. In the future work, we may be able to carry out in vivo PLIM study on animals after the probe optimization and achieving higher quality of sensing.

  1. AIE property of Pt1@block copolymer should be proved.

Response: Done. This suggestion was addressed in accordance with the plan proposed by the Reviewer 3 (“3. The fluorescence and phosphorescence quantum yields of the prepared particles should be provided, because they are essential data.” and “4. Please provide the non-normalized fluorescence spectra for AIE study”). Both experiments were performed (please refer to the responses #3 and #4 to the Reviewer 3 for more detail), and the results unambiguously support the proof of the AIE phenomenon: in the case of quantum yields, we observe the increase of this parameter from 2 % for 2.5 wt.% of Pt1 loading to 6 % for 12.4 wt.% of Pt1 loading. In the case of non-normalized luminescence spectra of Pt1@PCL-b-PEG measured in identical conditions, the steady increase of area under the spectrum at increased Pt1 loadings also quantitatively supports the conclusion on AIE.

Moderate English changes required.

Response: Done. In the course of the manuscript revision, we have checked all the text for typos and grammar and refreshed the least readable phrases to clarify them and improve the overall readability. All the changes can be tracked in the copy of the manuscript with tracking mode.

Reviewer 3 Report

 The authors prepared nano-particles based on Pt-complex and amphiphilic polymer, then investigated their application as lifetime intracellular oxygen biosensor. This work is new and the paper is well organized. Therefore, I suggest it be accepted after minor revision.

1.       The section of Materials and Methods takes too much space, and the details could be moved to SI.

2.       Besides DLS, TEM or SEM should be measured.

3.       The fluorescence and phosphorescence quantum yields of the prepared particles should be provided, because they are essential data.

4.       Please provide the non-normalized fluorescence spectra for AIE study.

5.       Please investigate the stability of the prepared particles. Is there any phase separation occurred?

Author Response

Reply to the Reviewer 3

Comments and Suggestions for Authors

The authors prepared nano-particles based on Pt-complex and amphiphilic  polymer, then investigated their application as lifetime intracellular  oxygen biosensor. This work is new and the paper is well organized. Therefore, I suggest it be accepted after minor revision.

  1. The section of Materials and Methods takes too much space, and the details could be moved to SI.

Response: Done. First, during the manuscript revision, Sections 2.1-2.9 were substantially rewritten to ensure the Editor’s request (”In addition, we detected high similarity/an overlap with previously published works in the Section 2 (Sectiones 2.1.-2.9.) of the manuscript. Please kindly rewrite the sections/parts during the revision stage.”). Second, Sections 2.1, 2.3 – 2.4., and 2.7. – 2.8. (all these sections repeat in general the methods developed and published earlier) were moved into the Supplementary Information. In the cases where it was possible, the overly detailed descriptions were substituted by citing the references of the preceding papers that have elaborated the same experiments/experimental setups. Now the sections being moved appear as Part 1 of Supplementary Material file.

  1. Besides DLS, TEM or SEM should be measured.

Response: In our work, TEM experiments were performed before the first submission, but the data obtained were not fully satisfying: i) we observed quite low amount of absorbed particles despite the variations in sample preparations; ii) disproportionately high portion of aggregates compared to micelles. Generally, we can see both micelles and aggregates in TEM pictures (micelles sizes are slightly higher compared to DLS data, most probably, due to flattening of nanoparticles during the adsorption on the support), and we provide these pictures as a material ‘for review only’ (pdf file, attached to this Reply), but we were unable to acquire a reliable statistics in these experiments and would prefer not to publish these data. Nevertheless, if the Reviewer will request to include these data into the manuscript, we will implement these TEM microphotographs into the Supplementary Information.

  1. The fluorescence and phosphorescence quantum yields of the prepared particles should be provided, because they are essential data.

Response: Done. We have measured quantum yields in aerated aqueous dispersions of Pt1@PCL-b-PEG with 2.5 wt.% and 12.4 wt.% of Pt1 loading. In these experiments, quantum yields increase from 2 to 6 %, respectively. In these experiments, an enhanced scattering was observed due to colloidal nature of the dispersions, so the measurement uncertainty was higher than that in conventional QY determinations in solution. We estimate it to be 20 to 30 %. Nevertheless, the result unambiguously proves the AIE phenomenon: we observe an increase of quantum yield from 2 % for 2.5 wt.% of Pt1 loading to 6 % for 12.4 wt.% of Pt1 loading.

We have added the following text to address the Reviewer’s suggestion:

Section 3.2., Page 7: “The quantum yield values measured in aerated aqueous dispersions of Pt1@PCL-b-PEG increased from 2 % for 2.5 wt.% of Pt1 loading to 6 % for 12.4 wt.% of Pt1 loading.”

Discussion, Page 13: “Figures 2, C and S4 demonstrate emergence and stepwise growth of a broad luminescence band centered at ca. 790 nm with the increase in Pt1 content in the micelles from 1.8 to 12.1 wt.%, accompanied by 3-fold increase in quantum yield values (from 2 % for 2.5 wt.% of Pt1 loading to 6 % for 12.4 wt.% of Pt1 loading).”

Supplementary Material file, Part 1, Section 4, Page 2: “Quantum Yield (QY) measurements were performed by a comparative method using the sample Pt1@PCL-b-PEG based on non-purified sample #2 of PCL-b-PEG (instead, the resulting micelles were purified from the precipitate by preparative centrifugation). Light emission diode was used as an excitation source and air-saturated solutions of [Ru(bpy)3][PF6]2 in water (QY = 0.040) was used as reference. In these experiments, an enhanced scattering was observed due to colloidal nature of the micellar dispersions. As a result, the measurement uncertainty appeared to be higher than that for conventional QY measurements in solutions; we estimate it to be 20 to 30 %.”

  1. Please provide the non-normalized fluorescence spectra for AIE study.

Response: Done. The non-normalized luminescence spectra are presented in Figure S4 in Supplementary material file. The following text was added into the Part 3.2. (Page 7) to address the Reviewer’s suggestion:

“Moreover, the overall emission intensity grows with increased loading of Pt1 into the micelles (Figure S4).”

  1. Please investigate the stability of the prepared particles. Is there any phase separation occurred?

Response: Done. The stability of the prepared particles has been described in detail in the section 3.1. (Page 6):

“The ‘empty’ and ‘loaded’ micelles prepared via the modified protocol were stable during the preparative centrifugation for 15 min at 20,000 rpm without any precipitation. Also, no phase separation or precipitation was observed at the concentrations up to 3 mg/mL, the upper concentration limit in stock solutions, as well as during dilution or storage at 4 °C for at least two weeks. Similar stability was observed in PBS solutions.”

English language and style are fine/minor spell check required

Response: Done. During the manuscript revision, we have checked all the text for typos and grammar and refreshed the least readable phrases to clarify them and improve the overall readability.
